# Cathepsin S Inhibition Suppresses Experimental Systemic Lupus Erythematosus-Associated Pulmonary Arterial Remodeling

**DOI:** 10.3390/ijms232012316

**Published:** 2022-10-14

**Authors:** Tzung-Hai Yen, Wan-Jing Ho, Yung-Hsin Yeh, Ying-Ju Lai

**Affiliations:** 1Department of Nephrology, Clinical Poison Center, Kidney Research Center, Center for Tissue Engineering, Chang Gung Memorial Hospital, Tao-Yuan 333, Taiwan; 2Cardiovascular Department, Chang Gung Memorial Hospital, Tao-Yuan 333, Taiwan; 3Department of Medicine, Chang Gung University College of Medicine, Tao-Yuan 333, Taiwan; 4Department of Respiratory Therapy, Chang Gung University College of Medicine, Tao-Yuan, 333, Taiwan; 5Department of Respiratory Care, Chang Gung University of Science and Technology, Tao-Yuan 333, Taiwan

**Keywords:** pulmonary arterial hypertension, lupus, animal model, cathepsin S, MRL/lpr

## Abstract

Patients with systemic lupus erythematosus (SLE) associated with pulmonary arterial hypnertension (PAH) receive targeted therapy for PAH to decrease pulmonary arterial systolic pressure and significantly prolong their survival. Cysteine cathepsin proteases play critical roles in the progression of cardiovascular disease. Inhibition of cathepsin S (Cat S) has been shown to improve SLE and lupus nephritis. However, the effect of Cat S inhibitors on SLE-associated PAH (SLE-PAH) remains unclear, and there is no animal model for translational research on SLE-PAH. We hypothesized that the inhibition of Cat S may affect PAH development and arterial remodeling associated with SLE. A female animal model of SLE-PAH, female MRL/lpr (Lupus), was used to evaluate the role of pulmonary arterial remodeling in SLE. The key finding of the research work is the establishment of an animal model of SLE associated with PAH in female MRL/lpr mice that is able to evaluate pulmonary arterial remodeling starting from the age of 11 weeks to 15 weeks. Cat S protein level was identified as a marker of experimental SLE. Pulmonary hypertension in female MRL/lpr (Lupus) mice was treated by administering the selective Cat S inhibitor Millipore-219393, which stimulated peroxisome proliferator-activated receptor-gamma (PPARγ) in the lungs to inhibit Cat S expression and pulmonary arterial remodeling. Studies provide an animal model of female MRL/lpr (Lupus) associated with PAH and a deeper understanding of the pathogenesis of SLE-PAH. The results may define the role of cathepsin S in preventing progressive and fatal SLE-PAH and provide approaches for therapeutic interventions in SLE-PAH.

## 1. Introduction

Pulmonary hypertension (PH) is defined as a mean pulmonary arterial pressure (mPAP) of ≥20 mmHg when measured by right heart catheterization [1]. Pulmonary arterial hypertension (PAH) is characterized by the narrowing and remodeling of the pulmonary arterial bed due to extensive endothelial dysfunction and adventitial and smooth muscle proliferation [2]. PAH is caused by genetic, environmental, and other predisposing conditions, including an imbalance in vasodilators and vasoconstrictors, inflammation, and an uncontrolled immune response [3,4]. These pathological changes result in increased vascular resistance and pulmonary artery (PA) pressure, which lead to right heart failure [1]. PH can be clinically classified into five groups that share similar pathological and hemodynamic characteristics and therapeutic approaches [5]. Group 1 PH encompasses idiopathic PAH (IPAH); heritable forms of PAH; drug- or toxin-induced PAH; and PAH associated with human immunodeficiency virus infection, portal hypertension, congenital heart disease, schistosomiasis, or chronic hemolytic anemia. PAH can also be related to various connective tissue diseases (CTDs), such as systemic sclerosis (SSc), systemic lupus erythematosus (SLE), rheumatoid arthritis (RA), and mixed connective tissue disease (MCTD) [5].

In patients with SLE, PAH is a fatal complication associated with a poor diagnosis and increased mortality [6,7,8,9]. The pooled prevalence estimate of PAH in SLE patients is 8% [10]. The various elements of SLE, from vasculitis and in situ thrombosis to interstitial pulmonary fibrosis, can lead to endothelial dysfunction, intimal hyperplasia, smooth muscle cell (SMC) proliferation, and medial thickening, which are similar to the changes seen in IPAH and PAH associated with scleroderma [8,11]. Unlike scleroderma, inflammation is a primary feature in SLE-associated PAH (SLE-PAH). Vasodilators, for example, phosphodiesterase 5 inhibitors, endothelin receptor antagonists, prostacyclin receptor agonists, and guanylate cyclase stimulators, have been used to treat patients with SLE-PAH [12]. Immunosuppressive, corticosteroid, and vasodilator therapies have shown much promise in treating SLE-PAH [13,14,15,16]. The clinical symptoms of PAH in SLE patients are non-specific, and the diagnosis could be delayed. The early detection of PAH in patients with CTD and the immediate initiation of intensive treatment are essential [6,16].

Major histocompatibility complex (MHC) class II-mediated priming of T and B lymphocytes is an essential concern regarding the autoimmunity observed in SLE and lupus nephritis. The cysteine protease cathepsin S (Cat S), a papain family member vital for the endolysosomal assembly of MHC II–antigen complexes, degrades the invariant peptide chain during the assembly of MHC II with antigenic peptides in antigen-presenting cells [17]. A Cat S inhibitor was shown to suppress MHC class II-mediated CD4 T-cell and B-cell priming and thus improve SLE and lupus nephritis in a mouse model of SLE (MRL/lpr) [18]. Cysteine cathepsin proteases also play critical roles in the progression of cardiovascular disease [19]. Cathepsins are present in lysosomes of various types of vascular cells, including endothelial cells, vascular smooth muscle cells (VSMCs), and macrophages in the cardiovascular disease [20,21]. Our previous study showed the pathophysiological significance of Cat S activity in pulmonary vascular remodeling due to the disruption of peroxisome proliferator-activated receptor-gamma (PPARγ) in the lungs of IPAH patients and in pulmonary arterial smooth muscle cells (PASMCs) of the rat MCT-induced PAH model [22]. In addition, PAH in a rat model of MCT-PAH was treated by the Cat S inhibitor Millipore-219393, which stimulates PPARγ to inhibit Cat S expression, thus suppressing the proliferation and migration of PASMCs to reduced RVSP in this model [22].

In recent works on the latest development of the research topic, pulmonary hypertension (PH) is defined as a mean pulmonary arterial pressure (mPAP) of ≥20 mmHg [1,5]. PAH associated with systemic lupus erythematosus is classifies into WHO group 1 [5]. As the symptoms of PAH in SLE can be mild and non-specific, in the early stages, that led to poor diagnosis and increased mortality. Recently, Pan Y and colleagues examined the predictive factors of PAH in SLE patients by a chart review study and found that interstitial pneumonitis, myocardial damage, and high IgG were predictive factors of PAH in SLE patients [23]. In addition, baseline serositis, 6 min walking distance, and cardiac index could be recognized as independent prognostic factors of the treatment goal achievement (TGA) for the patients of SLE-PAH [16]. Cat S inhibitors were shown to suppress MHC class II-mediated CD4 T-cell and B-cell priming and thus improve SLE and lupus nephritis in a mouse model of SLE (MRL/lpr) [18]. Overexpression of cathepsins has been reported to exacerbate lupus pathogenesis [24] and cardiovascular diseases [19,21,22]. Inhibition of Cat S is predicted to suppress antigen presentation via MHC class II, T-cell, and B-cell activation and antibody production by B cells; thus, Cat S is an attractive therapeutic target for SLE [18,25], but the potential effect on SLE-PAH is not clear.

Cathepsins, potent elastases that can damage elastin, are secreted by many types of cells, including pulmonary vascular endothelial cells, smooth muscle cells, and macrophages [4,22,26]. Our previous study demonstrated that Cat S overexpression induces pulmonary arterial remodeling [22]. In a rat model, PAH can be treated with the selective Cat S inhibitor Millipore-219393, which stimulates PPARγ to inhibit Cat S expression [22]. However, the effect of Cat S inhibitors on SLE-associated PAH (SLE-PAH) remains unclear. Moreover, there is no report on understanding the animal model for translational research of SLE-PAH. In this study, we aimed to observe and establish an animal model of SLE-associated PAH from weeks 11 to weeks 31, female MRL/lpr (Lupus), to evaluate the role of pulmonary arterial remodeling in SLE, and the mechanisms underlying the suppressive effects of Cat S inhibitors on vascular remodeling were analyzed by α-SMA staining to characterize medial vascular hypertrophy in SLE associated with PAH (SLE-PAH). In addition, the effects on the molecular level of PPARγ were analyzed by immunohistochemistry and immunoblotting to provide mechanistic explanation of Cat S inhibitor-induced changes.

## 2. Results

### 2.1. Cathepsin S Protein Level Is a Marker of Experimental SLE

To investigate the specific expression pattern of Cat S in the lungs of experimental SLE models, we compared organ-specific Cat S protein levels in female MRL/lpr (SLE) mice and male MRL/lpr (Con) mice. We found elevated Cat S protein levels in lung, kidney, spleen, and liver tissues from female MRL/lpr (SLE) mice compared to male MRL/lpr (Con) mice. We also observed decreased protein levels of PPARγ, a downstream effector of Cat S signaling, in the lungs of female MRL/lpr (SLE) mice (Figure 1A,B). Quantification of protein levels by Western blot analysis revealed a significant increase in Cat S (24 kDa) in female experimental mice compared to male control mice (Figure 1B), along with significant decreases in PPARγ protein levels (Figure 1C) in the lung, kidney, spleen, and liver. To determine whether Cat S overexpression plays a role in the development of PAH, we studied the development of PH in female MRL/lpr (SLE) mice. These female mice with SLE-induced PH exhibited increased RVSP compared with male MRL/lpr (Con) mice (Figure 1D). Our results indicate that Cat S and PPARγ levels correlate with the severity of PAH in a murine model of MRL/lpr (SLE).

### 2.2. Female MRL/lpr (SLE) Mice Show Pulmonary Arterial Remodeling and Right Ventricular Hypertrophy

The MRL/lpr mouse is a standard model of lupus nephritis [27,28]. The lupus-like syndrome in MRL/lpr mice has an earlier onset and is more acute in females than in males. In response to renal immune injury, monocytes, T lymphocytes, and neutrophils infiltrate the kidney and mediate proteinuria, tissue injury, and renal dysfunction. However, PAH and vascular remodeling are never observed in female MRL/lpr (SLE) mice. Moreover, the pathophysiological pathways linking PAH to SLE have not been established, and there is no animal model for translational research on PAH-SLE. To determine whether and when female MRL/lpr (SLE) mice develop PAH, we observed the pathology of lung tissue and studied the type of muscular PA remodeling. The percent MWT determined by α-SMA staining was utilized to characterize medial vascular hypertrophy, and we analyzed the RV ratio to identify right ventricular hypertrophy, the extent of which is expressed as the ratio of the RV wall weight to the combined weight of the free LV wall and ventricular septum. Female MRL/lpr (SLE) mice exhibited worse right ventricular hypertrophy and vascular remodeling than male MRL/lpr (Con) mice after 11 weeks (Figure 2A,B). Therefore, female MRL/lpr (SLE) mice developed right ventricular hypertrophy and showed luminal narrowing of small PAs, consistent with advanced PAH.

### 2.3. Female MRL/lpr Mice Show Partial Muscularization of the Pulmonary Arteries

To study the pathology of the pulmonary vasculature in female MRL/lpr (SLE) mice, we quantitatively assessed the degree of muscularization of PAs with diameters between 25 and 50 μm. In male MRL/lpr (Con) mice, most vessels of this diameter were commonly nonmuscularized, as shown in Figure 3A. The degree of muscularization of pulmonary arteries in MRL/lpr mice is demonstrated by smooth muscle actin (SMA) (brown) staining for identifying vascular SMCs, whereas female MRL/lpr (SLE) mice showed a significant increase in partially and fully muscularized arteries at weeks 11 and 15 and a significant increase in the percentage of partially muscularized PAs. In female MRL/lpr (SLE) mice, a dramatic decrease in nonmuscularized PAs was observed at weeks 11 and 15 (Figure 3B), with a concomitant increase in partially muscularized and fully muscularized PAs (Figure 2A and Figure 3B).

### 2.4. Cat S Inhibitor Upregulates PPARγ and Suppresses Cat S Expression

Inhibition of Cat S is predicted to suppress antigen presentation via MHC class II, T-cell, and B-cell activation, and antibody production by B cells; thus, Cat S is an attractive therapeutic target for SLE [18,25], but the potential effect on SLE-PAH is not clear. Cathepsins, potent elastases that can damage elastin, are secreted by many types of cells, including vascular endothelial cells, VSMCs, and macrophages [4,22]. Our previous study demonstrated that Cat S overexpression induces pulmonary arterial remodeling [22]. In a rat model, PAH can be treated with the selective Cat S inhibitor Millipore-219393, which stimulates PPARγ to inhibit Cat S expression [22]. Therefore, we hypothesized that Millipore-219393 might regulate Cat S expression and consequently mediate the development of PAH via PPARγ. First, we analyzed PPARγ and Cat S expression levels in lung tissues of SLE mice. Western blot analysis showed that Cat S protein levels were increased in female MRL/lpr (SLE) mice compared with male MRL/lpr (Con) mice (Figure 4A). Conversely, PPARγ protein levels in lung tissues were lower in female MRL/lpr (SLE) mice than in male MRL/lpr (Con) mice (Figure 4A). In mice treated with 5 µg/100 g/d Millipore-219393, PPARγ expression was specifically increased, and Cat S expression was decreased in lung tissue (Figure 4A). Next, the efficacy of Millipore-219393 was evaluated in lung tissue lysates and serum samples from MRL/lpr mice. As expected, Cat S activity was increased in female MRL/lpr (SLE) mice at 15 weeks compared with male MRL/lpr mice (Figure 4B,C). Administration of the Cat S inhibitor Millipore-219393 from the age of 11 weeks to 15 weeks reduced Cat S activity in female MRL/lpr (SLE) mice (Figure 4B,C). These findings suggest that Millipore-219393 not only inhibited Cat S but also upregulated PPARγ expression in the lungs of the experimental SLE (MRL/lpr) mouse model.

### 2.5. Cat S Inhibitor Upregulates PPARγ and Suppresses Cat S Expression to Prevent the Pulmonary Arterial Remodeling and Right Ventricular Hypertrophy in Experimental SLE

To determine whether increased Cat S expression is involved in PA remodeling in experimental SLE (female MRL/lpr) mice, we treated female MRL/lpr mice with 5 µg/100 g/day Millipore-219393 or vehicle via ip injection from the age of 11 weeks to 15 weeks. The efficacy of Millipore-219393 was confirmed based on the analysis of serum Cat S activity (Figure 4B,C). Immunohistochemical staining showed that Millipore-219393 treatment significantly reduced Cat S expression (Figure 5A) and increased PPARγ expression (Figure 5B) in the PAs of female MRL/lpr mice. We previously showed that Cat S plays a significant role in the development of vasculopathy in PAH-related SLE, which can be prevented by the Cat S inhibitor Millipore-219393. Accordingly, we determined whether treatment with this Cat S inhibitor effectively prevents pulmonary arterial remodeling in female MRL/lpr mice and quantified the degree of muscularization of PAs with a diameter ranging from 25 to 100 µm by measuring the MWT. As expected, the medial wall of PAs was significantly thicker in female MRL/lpr (SLE) mice than in male MRL/lpr (Con) mice. Millipore-219393 treatment of female MRL/lpr (SLE) mice significantly reduced the MWT of the PA (Figure 5C) compared with that in vehicle-treated MRL/lpr (SLE) mice. Our findings demonstrate that this Cat S inhibitor effectively prevents the development of PAH in female MRL/lpr (SLE) mice.

## 3. Discussion

The novel finding of the present study is the establishment of an animal model of SLE associated with PAH in female MRL/lpr mice that is able to evaluate pulmonary arterial remodeling from the age of 11 weeks to 15 weeks. Cat S protein level was identified as a marker of experimental SLE, and female MRL/lpr (SLE) mice with PH exhibited increased RVSP compared with male MRL/lpr (Con) mice. The Cat S inhibitor Millipore-219393 was found to be an effective treatment in this established experimental model of SLE associated with PAH. The treatment effects included reductions in right ventricular hypertrophy and pulmonary arterial remodeling in female MRL/lpr (SLE) mice. Moreover, the importance of Cat S and PPARγ in the development of PH-associated SLE was proven by experiments using a selective inhibitor of Cat S (Millipore-219393), which upregulated PPARγ and suppressed Cat S expression to prevent pulmonary arterial remodeling and right ventricular hypertrophy in experimental SLE.

SLE is a systemic autoimmune disease with immune complex-related manifestations characterized by the presence of pathogenic autoantibodies to nuclear antigens and elevated levels of immune complexes, among other immunological abnormalities such as the generation of autoreactive B and T cells, reflecting a global loss of self-tolerance [29,30]. Recent research has identified Cat S as a potentially attractive therapeutic target for the clinical treatment of SLE through its inhibition of antigen presentation. The plasma Cat S level may be a marker for the stratification of patients with SLE for Cat S inhibitor therapy (ASP1617 and RO5461111) [18,25]. The critical role of MHC class II-mediated T-cell and B-cell priming for autoimmunity and the nonredundant roles of Cat S in peptide loading and MHC class II assembly prompted speculation that Cat S is a potential therapeutic target in SLE [18,25]. In 2009, Rupanagudi KV and colleagues tested the Cat S inhibitor RO5461111 in the MRL-Fas (lpr) (SLE) mouse model. The experimental protocol started at the age of 12 weeks, with the administration of a medicated diet formulated with a Cat S inhibitor (RO5461111, 262.5 mg/kg chow). The mice consumed an approximate dose of 1.31 mg per day. At the end of week 20, the mice were sacrificed, and the experimental results were obtained [18]. When given to female MRL-Fas (lpr) (SLE) mice with lupus nephritis, RO5461111 significantly reduced the activation of spleen dendritic cells and the subsequent expansion and activation of CD4 T cells and CD4/CD8 double-negative T cells. Cat S inhibition by RO5461111 suppressed follicular B-cell maturation to plasma cells and Ig class switching and reduced hypergammaglobulinemia and lupus autoantibody production in MRL-(Fas) lpr mice. RO546111 also improved lupus nephritis, protecting kidneys and reducing lung inflammation, as indicated by lower mRNA levels of inflammatory cytokines and fewer Mac-2+ macrophages [18]. However, autoimmune markers are the main therapeutic targets in all these approaches; although the idea has been hypothesized, there is little proof (either experimental or clinical) that Cat S inhibitors have the direct ability to reverse the vascular and muscular changes in SLE associated with PAH.

RO5461111 is not clinically available. Therefore, Millipore-219393, a selective Cat S inhibitor, was chosen to test our hypothesis that inhibiting Cat S may affect the development of PAH and arterial remodeling associated with SLE. Millipore-219393 has been utilized in studies of the contribution of autophagy to anti-TNF-induced macrophages [31], and the neuroinflammation [32]. However, to date, no interventional data are available on the therapeutic use of Millipore-219393 in experimental or clinical SLE. In our previous study, we showed overexpression of Cat S and degradation of elastic laminae in the lungs of IPAH patients and in the PASMCs of the MCT-induced PAH rat model. In addition, Millipore-219393-treated PH in MCT-PAH rats; this compound stimulated PPARγ to inhibit Cat S expression, thus suppressing the proliferation and migration of MCT-PAH PASMCs to reduce RVSP [22]. Herein, we demonstrate a mechanistic link between Cat S signaling and PPARγ, and the results suggest that PPARγ is upstream of Cat S signaling. Furthermore, we showed that Millipore-219393 can markedly reverse PAH in a rat model of MCT-induced PAH [22]. Treatment did not affect systemic arterial pressure, proving the selectivity of this approach for abnormal pulmonary circulation and providing evidence for the absence of important anti-pulmonary arterial remodeling effects. In this study, we established an animal model of SLE-associated PAH in female MRL/lpr mice aged from 11 to 15 weeks to evaluate the role of pulmonary arterial remodeling in SLE, as seen in Figure 2 and Figure 3. The Cat S inhibitor Millipore-219393 was found to be an effective treatment in this established experimental model of SLE-associated PAH from week 11 to 15. The treatment effects in female MRL/lpr (SLE) mice included reduced right ventricular hypertrophy and less pulmonary arterial remodeling, as seen in Figure 4 and Figure 5.

Abnormal expression and activity of Cat S, a lysosomal protease in the cysteine cathepsin protease family, is related to the pathogenesis of various diseases and to an important intracellular role in MHC class II antigen presentation in autoimmune diseases such as SLE [18,33]. Alterations in Cat S activity are associated with pulmonary disease, cancer, cardiovascular disease, and diabetes [34]. Cardiovascular research studies have shown that vascular cells associated with atherosclerotic lesions overexpress Cat S in the absence of a change in cystatin C expression, suggesting a shift in the balance between cathepsins and their inhibitor that favors cardiovascular wall remodeling [35,36]. In our previous study, increases in Cat S expression and activity were shown to correlate with disease severity in experimental MCT-induced PAH and in patients with IPAH [22]. Changes in vasculature structure and function are essential in the pathogenesis of vascular diseases, including PAH, atherosclerosis, and restenosis [19]. We showed that PAH vasculopathy is characterized by high steady-state levels of Cat S and degradation of elastin lamina in the SMCs lining the proximal and distal PAs in both patients with PH and a rodent model [22]. Moreover, Millipore-219393 decreased Cat S levels in lung and serum samples and upregulated PPARγ, indicating that it may be a PPARγ agonist, as seen in Figure 4 and Figure 5.

PPARγ acts as a vasoprotective metabolic regulator in SMCs and endothelial cells [37]. Loss of PPARγ has been linked to the development of severe PAH [38]. In addition, activation of PPARγ signaling can attenuate PH in experimental models [20]). We confirmed that PPARγ knockdown led to increased Cat S expression in cultured PASMCs [22]. PPPARγ is a member of the nuclear receptor superfamily of ligand-activated transcription factors and regulates adipogenesis and metabolism by binding to PPAR response elements (PPREs) in the promoter regions of various target genes related to vascular cell proliferation and migration [39]. Therefore, PPARγ is a potential therapeutic target for transcriptional regulation of VSMC proliferation and migration in cardiovascular diseases [40,41]. We demonstrated that PPARγ was downregulated in the lung tissues of MCT-PAH rats and confirmed that PPARγ knockdown led to increased Cat S expression in cultured PASMCs [22]. Previously, we found that the selective Cat S inhibitor Millipore-219393 not only inhibits Cat S activity but also acts as a PPARγ agonist that induces PPARγ expression in PASMCs and lung tissue [22]. In addition, we demonstrated that Millipore-219393 treatment effectively prevents increases in RVSP, RV weight, and PA muscularization in rats challenged with MCT, suggesting that Millipore-219393 has potential therapeutic effects in PAH [22]. However, few studies have focused on cysteine endoproteases in PAH-related SLE. Direct evidence for the involvement of the PPARγ–Cat S signaling pathway in the development of SLE-induced PH and the therapeutic effects of Cat S inhibition was derived from the following results: (a) Cat S expression was markedly upregulated in the lungs and serum of female MRL/lpr (SLE) mice, which is consistent with previous data showing that Cat S levels are significantly increased, but as is PPARγ level are decreased in the PAs of MCT-treated rats [22]; (b) this increased Cat S expression was reversed by Millipore-219393 treatment, in parallel with beneficial effects on pulmonary arterial remodeling and right ventricular hypertrophy; and (c) Cat S levels were increased in female MRL/lpr (SLE) mice but decreased in response to Millipore-219393 treatment and the induction of PPARγ. Increased expression of PPARγ has been reported to be protective in the pathogenesis of SLE [42,43]. In addition, the PPARγ agonist pioglitazone modulates abnormal T-cell responses in SLE [44]. Another PPARγ agonist, rosiglitazone, decreases autoantibody production and prevents atherosclerosis and renal diseases in mouse models of SLE by upregulating adiponectin [45]. The importance of Cat S and PPARγ in the development of SLE associated with PAH was proven by experiments using the selective Cat S inhibitor Millipore-219393, which upregulated PPARγ and suppressed Cat S expression to prevent pulmonary arterial remodeling and right ventricular hypertrophy in experimental SLE.

The novel finding of the present study is the establishment of an animal model of SLE associated with PAH in female MRL/lpr mice that are able to evaluate pulmonary arterial remodeling from the age of 11 weeks to 15 weeks. Our study is the first to describe the successful therapeutic use of a Cat S inhibitor in an animal model of SLE associated with PAH. We demonstrated that treatment with Millipore-219393 reversed the right ventricular hypertrophy and vascular structural changes in the SLE animal model. Furthermore, this study provides evidence at the cellular and molecular levels for the involvement of Cat S-PPARγ in disease development and treatment response in lung tissue. Nevertheless, this laboratory research is limited by a lack of clinical research. Further studies are warranted. Taking into account the limitations of experimental studies, we believe that our findings will encourage consideration of this novel therapeutic approach for patients with SLE associated with PAH, in which Cat S may be markedly upregulated, as suggested by recent findings in SLE patients. Clinical studies revealed that the circulating levels of Cat S increased in chronic kidney patients with cardiovascular disease [46], and the levels also increased with chronic kidney disease (CKD) progression [47]. This is important, because chronic kidney disease is a frequent finding in SLE patients and prediction of grave outcomes [48]. More than half of SLE patients develop lupus nephritis and 20% eventually progress to end-stage kidney disease. As several Cat S inhibitors have been evaluated as anti-SLE drugs in experimental studies, this approach may be developed and applied in the clinic in the future as a new therapy for SLE associated with PAH.

## 4. Materials and Methods

### 4.1. Mice and Experimental Protocol

The mouse strain MRL/MpJ-Faslpr/2J (MRL/lpr) was purchased from Jackson Laboratory, USA, and the mice were maintained at the national Laboratory Animal Center, Tainan, Taiwan, and the Chang Gung University Laboratory Animal Center [27]. The experimental mice received once-daily intraperitoneal (ip) injections of the Cat S inhibitor Millipore-219393 (Calbiochem, Merck, Darmstadt, Germany) at a dosage of 5 µg/100 g/day or isotonic saline as a vehicle control. Mice were examined after four weeks of treatment. The Cat S inhibitor Millipore-219393 was chosen to target Cat S signaling in female MRL/lpr (Lupus) mice, and the dose was calculated based on published studies [22,49]. We used the ARRIVE1 reporting guidelines [25]. All animal experiments were reviewed and the study protocols approved by the Institutional Animal Care and Use Committee (IACUC) of Chang Gung University (permit number: CGU14-149). Mice were anesthetized with a combination of Zoletil (20 mg/kg) (Virbac, Carros, France) and Rompun (5 mg/kg) (Bayer, Leverkusen, Germany) by intraperitoneal injection, and all efforts were made to minimize suffering. Housing and maintenance was provided by Chang Gung University; all animals were fed a standard chow diet with free access to water.

### 4.2. Hemodynamic Measurements and Cardiovascular Evaluation

Hemodynamic measurements were performed in mice at age 15 weeks, as described previously [50]. To monitor hemodynamics, animals were deeply anesthetized as previously described [50]. A 1.6F catheter-tipped pressure transducer (Transonic Scisense, London, ON, Canada) was inserted through the right jugular vein to measure the right ventricular systolic pressure (RVSP) [50]. To assess right ventricular hypertrophy, the RV was detached from the left ventricle (LV) wall and septum (LV + S), and the weight of the RV, free LV wall, and ventricular septum was determined. RV hypertrophy was stated in the ratio of weight of the RV wall and that of the free LV wall plus the ventricular septum (LV + S) [50].

### 4.3. Immunohistochemical Analysis

Immunohistochemical analysis of lung tissues was performed with primary antibodies against α-smooth muscle actin (α-SMA) (Sigma-Aldrich, Burlington, MA, USA; 1:1000), Cat S (Abcam, Cambridge, UK; 1:200), and PPARγ (Santa Cruz, Dallas, TX, USA; 1:200). Staining of α-SMA was used to indicate the medial layer of small PAs for the assessment of medial wall thickness (MWT). To evaluate Cat S signaling and exclude autofluorescence from elastic fibers, we exposed the tissue samples to blocking solution instead of the primary antibody as a negative control. To locate protein expression, lung tissue sections were incubated with rabbit anti-Cat S and mouse anti-α-SMA antibodies for 1 h and then with Alexa-488-conjugated secondary antibody (for Cat S; green, Invitrogen, Waltham, MA, USA; 1:500) or Cy3-conjugated secondary antibody (for α-SMA; red, Invitrogen, Waltham, MA, USA; 1:500) at room temperature for 30 min and observed with a confocal microscope (Confocal TCS SP8XL; Leica, Wetzlar and Mannheim, Germany) at the Microscope Core Laboratory of Chang Gung Memorial Hospital.

### 4.4. Assessment of Media Wall Thickness (MWT)

The percent variability in media wall thickness (MWT) was utilized to characterize medial vascular hypertrophy and vascular remodeling seen in PAH after α-SMA staining. Under 200× magnification, the MWT was determined as the distance between the internal and external elastic laminae and the wall thickness of a pulmonary vessel by measuring the lamina elastica externa and lumen in two perpendicular directions in vessels between 50 to 100 μm encountered per slide using ImageJ software (http://rsb.info.nih.gov/ij/download.html; accessed on 12 May 2022). For vascular sections, the diameter was calculated as (longest diameter + shortest diameter)/2 [22].

### 4.5. Western Blot Analysis

Lung tissue samples were homogenized in lysis buffer as described previously [22]. The membranes were probed with 1 of the following antibodies: anti-Cat S (Santa Cruz, Dallas, TX, USA; 1:2000) and anti-PPARγ (Santa Cruz, Dallas, TX, USA; 1:2000) antibodies or anti-GAPDH (Santa Cruz, Dallas, TX, USA; 1:10,000) loading control. Secondary antibodies specific for peroxidase-conjugated anti-mouse IgG (Sigma-Aldrich; 1:10,000) or anti-rabbit IgG (Sigma-Aldrich, Burlington, MA, USA; 1:5000) were used as needed. Bound antibodies were detected by chemiluminescence with the use of an enhanced chemiluminescence (ECL) detection system (Amersham Biosciences, Amersham, UK). Sample bands were normalized to GAPDH and quantified by densitometry.

### 4.6. Cathepsin S Activity Assay

Cat S activity from the homogenized lung tissue and serum was assessed using a fluorescence-based activity assay kit (Biovision Research Product, Waltham, MA, USA) in which the preferred Cat S substrate is labeled with amino-4-trifluoromethyl coumarin, and the results were processed according to the manufacturer’s instructions as described previously [22].

### 4.7. Statistical Analysis

For multiple groups, one-way ANOVA with the post hoc Bonferroni test was used to compare data among groups. Data are reported as the mean and standard error of the mean (SEM). Unpaired *t*-tests were used to determine differences between two groups. A value of *p* ≤ 0.05 was considered to indicate statistical significance.

## 5. Conclusions

In patients with SLE, PAH is a fatal complication associated with a poor diagnosis and increased mortality. The clinical symptoms of PAH in SLE patients are non-specific and the diagnosis could be delayed. The early detection of PAH in patients with CTD and the immediate initiation of intensive treatment are essential. The novel finding of the present study is the establishment of an animal model of SLE associated with PAH in female MRL/lpr mice that is able evaluate pulmonary arterial re-modeling starting from the age of 11 weeks to 15 weeks. The Cat S protein level was identified as a marker of experimental SLE and female MRL/lpr (SLE) mice. The Cat S inhibitor Millipore-219393 was found to be an effective treatment in this established experimental model of SLE associated with PAH. The treatment effects included reductions in right ventricular hypertrophy and pulmonary arterial remodeling in female MRL/lpr (SLE) mice. Moreover, the importance of Cat S and PPARγ in the development of PH-associated SLE was proven by experiments using a selective inhibitor of Cat S (Millipore-219393), which upregulated PPARγ and suppressed Cat S expression to prevent pulmonary arterial remodeling and right ventricular hypertrophy in experimental SLE.

## Figures and Tables

**Figure 1 ijms-23-12316-f001:**
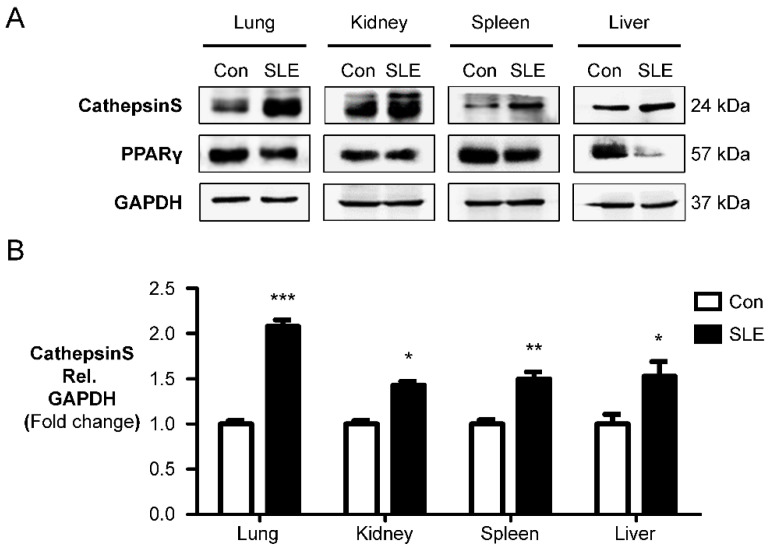
Increased cathepsin S and decreased PPARγ expression are the markers of PAH related to SLE. (**A**) Western blot analysis of cathepsin S and PPARγ in the organs of female MRL/lpr mice (SLE) compared to male MRL/lpr (Con) mice. (**B**,**C**) Relative expression values obtained by densitometry of the cathepsin S and PPARγ protein normalized to GAPDH (*n* = 3 per group). GAPDH = glyceraldehyde-3-phosphate dehydrogenase. The data are presented as the mean ± SEM of four samples in each group. * *p* < 0.05, ** *p* < 0.01, *** *p* < 0.001 compared with the control (Con) group. (**D**). Representative RVSP tracings of male and female MRL/lpr mice at the age of 15 weeks. Pulmonary hypertension (indicated by an elevated RVSP) showed increased RVSPs in the female MRL/lpr mice (SLE) groups. Increased cathepsin S and loss PPARγ protein expression enhanced the RVSP in female MRL/lpr mice (SLE). The data are presented as the mean ± SEM of four samples in each group. *** *p* < 0.001 compared with the Con group.

**Figure 2 ijms-23-12316-f002:**
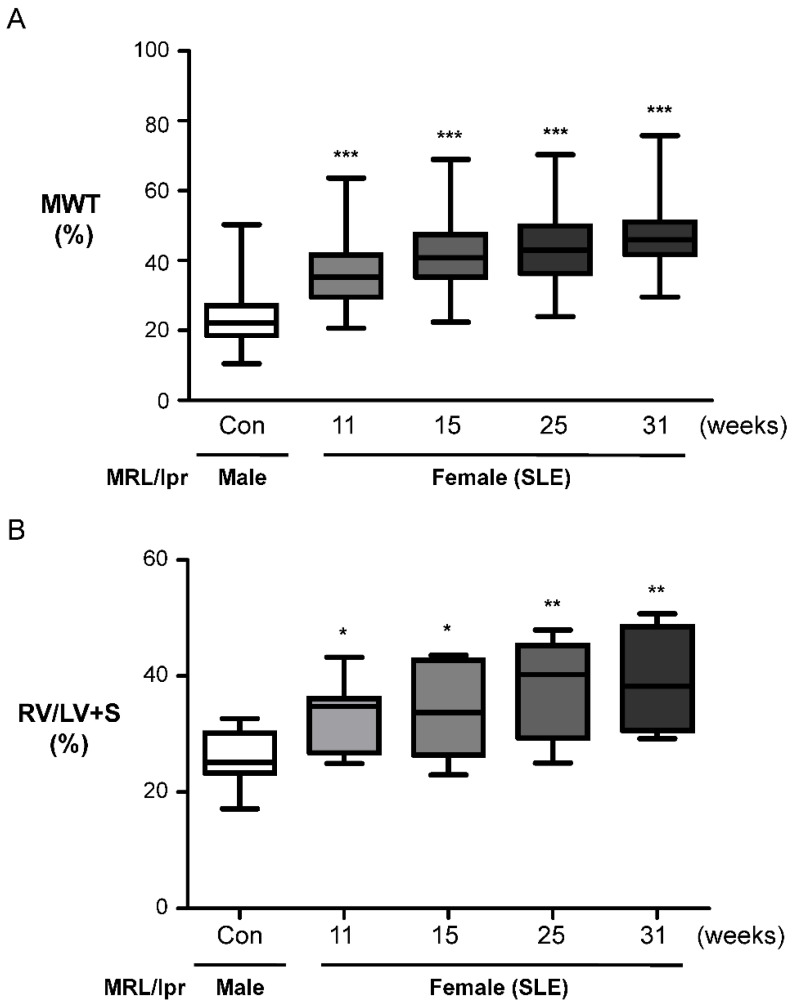
Female MRL/lpr mice showed increased development of pulmonary arterial remodeling and right ventricle hypertrophy. (**A**) Media wall thickness (MWT) values of small PAs (50–100 µm) identified by SMA staining. The data are presented as the mean ± SEM (*n* = 3–5). *** *p* < 0.001 versus male MRL/lpr mice (Con); one-way ANOVA. (**B**) The ratios of RV to LV plus the septum weight (RV/LV+S) in 5 different groups are shown for male MRL/lpr mice and female MRL/lpr mice. The data are presented as the mean ± SEM (*n* = 3–5), * *p* < 0.05 and ** *p* < 0.01 compared with male MRL/lpr mice. one-way ANOVA.

**Figure 3 ijms-23-12316-f003:**
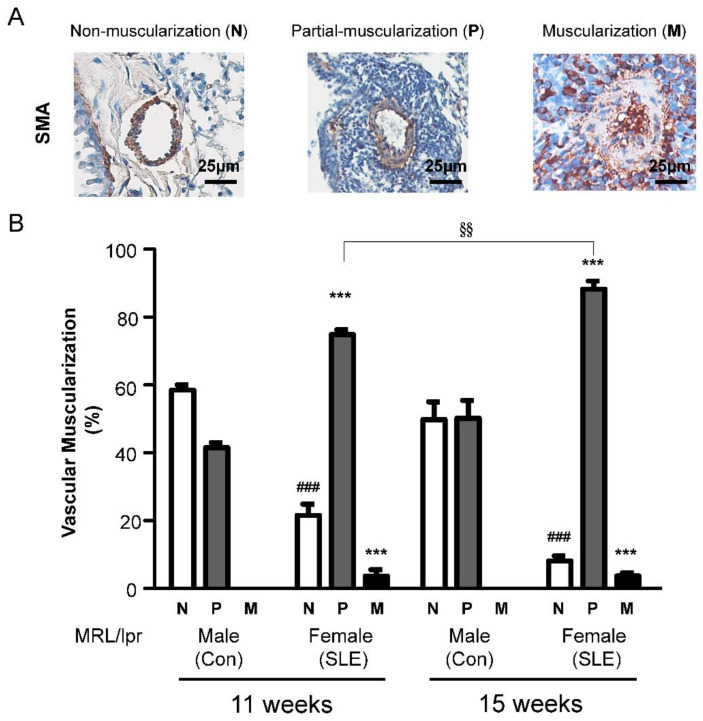
The degree of muscularization of pulmonary arteries in male and female MRL/lpr mice. (**A**) Degree of non (N), partial (P), or full (M) muscularization. (**B**) Percentage of non (N)-, partially (P)-, or fully (M)-muscularized pulmonary arteries, as a percentage of total pulmonary arteries (size 25–100 µm) in a cross section. A total of 60–80 intra-acinar vessels was analyzed in each lung. The results from 11-week and 15-week groups (Con mice and SLE mice) are presented. The data are presented as the mean ± SE. *** *p* <0.001 versus the Con group; ### *p* < 0.001 versus the Con group, showed a significantly different increase or decrease from the Con group. §§ *p* < 0.01 versus the 11-week SLE group. ANOVA with Bonferroni’s post hoc test.

**Figure 4 ijms-23-12316-f004:**
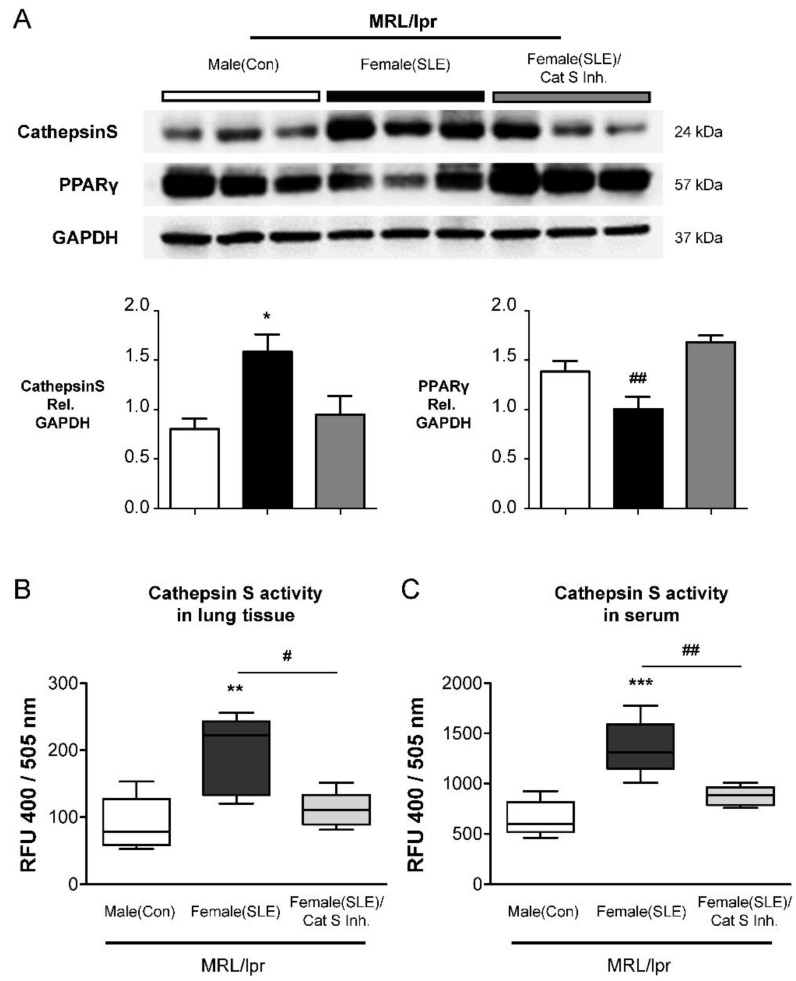
Effect of Cat S inhibitor on PPARγ stimulation, and decreasing Cat S expression in the lungs and circulation of female MRL/lpr mice. (**A**) Representative immunoblotting densitometry quantification of protein expression is indicated for the lungs of the three groups. Cat S (24 kDa band) and PPARγ (57 kDa band). The bars represent the mean ± SEM for *n* = 3 samples. * *p* < 0.05 (increase) or ## *p* < 0.05 (decrease) compared with the male MRL/lpr mice (Con) group. (**B**) Cat S activity in the lung tissue lysate. (**C**) Cat S activity in the serum. Each value (mean ± SE (*n* = 6–7) is expressed. ** *p* < 0.01, *** *p* < 0.001 (increase) versus male MRL/lpr (Con) mice, or # *p* < 0.05, ## *p* < 0.01 (decrease) compared with the female MRL/lpr mice, one-way ANOVA, Bonferroni posttest.

**Figure 5 ijms-23-12316-f005:**
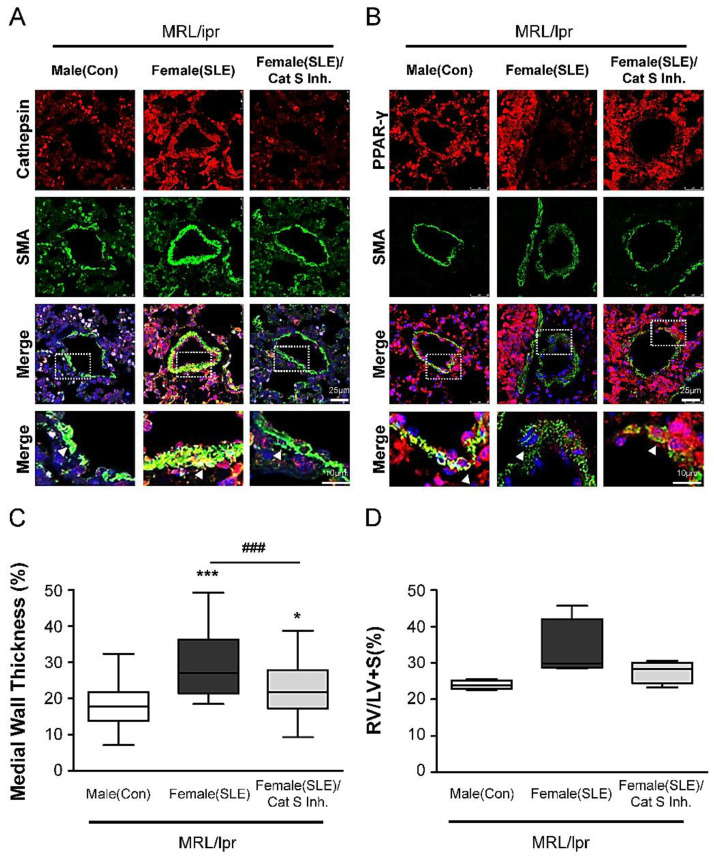
Effect of Cat S inhibitor in preventing the development of pulmonary arterial remodeling and right ventricle hypertrophy. (**A**) Immunohistochemistry analysis of SMA with cathepsin S, and (**B**) SMA with PPARγ in the pulmonary vascularity (scale bar: 25 µm). (**C**) Medial wall thickness (MWT) of small PAs identified by SMA staining. The degree of MWT was compared among 3 groups. (**D**) The ratio of RV to LV plus septum weight (RV/LV+S) is shown. Each value (mean ± SEM (*n* = 3–5)) is expressed. * *p* < 0.05, *** *p* < 0.001 versus male MRL/lpr mice (Con) group or ### *p* < 0.001 compared with the female MRL/lpr mice (SLE) group, one-way ANOVA, Bonferroni posttest.

## Data Availability

Not applicable.

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
