# Peer review of "Cathepsin S Inhibition Suppresses Experimental Systemic Lupus Erythematosus-Associated Pulmonary Arterial Remodeling"

_ijms, 2022, doi:10.3390/ijms232012316_

Round 1

Reviewer 1 Report

- I'd like to congratulate the authors. It is a very original and significant work that will be interesting to the readers.

- PAH is a rare complication in SLE although fatal in the majority of cases. The authors used an animal model of 188 SLE associated with PAH in female MRL/lpr mice to evaluate pulmonary arterial remodeling from the age of 11 weeks to 15 weeks. They show that Cat S and PPARγ levels correlate with the severity 97 of PAH in a murine model of MRL/lpr (SLE).

- Recent research showed that Cat S is an therapeutic target for SLE through its inhibition of antigen presentation. The authors showed that by inhibiting  RO5461111, follicular B-cell maturation to plasma cells was suppressed, resulting in reduced hypergammaglobulinemia and lupus autoantibody production in MRL-(Fas) lpr mice. 

- I think the order of the elements is not right. The discussion should come after materials and methods and not before.

- The authors should include a statement about ethics and whether this project passed the ethical committee of their center. 

Reviewer 2 Report

There are various key comments to be addressed by authors. A major revision is recommended.
Comment 1. Abstract:
(a) Define the acronym PAH.
(b) Share the key findings/results of the research work.
Comment 2. Keywords, more terms should be included to better reflect the scopes of the paper.
Comment 3. Section 1 Introduction:
(a) Enhance the discussion on the importance of the research topic.
(b) Ensure proper spacing “SLE-PAH[12]”. Check carefully the rest of the contents.
(c) Enhance the literature review by summarizing the methodology, results, and limitations of the existing works.
(d) Summarize the research contributions of the paper.
Comment 4. Methodology should be presented before the results.
Comment 5. Section 2 Results:
(a) Refer to the format for callout and caption of the figures.

(b) Ensure high resolutions for all figure.
(c) The procedures for the experimental studies should be presented in detail.
(d) Avoid presenting the explanation in the captions of the figures. Move the contents back to the main-text.
(e) Performance comparison with existing works is expected.
Comment 6. Section 3 Discussion:
(a) It seems that some of the contents refer to the results. Please include the callouts for the figures.
Comment 7. Section 4 Materials and methods:
(a) Poor organization is found in this section.
Comment 8. Conclusion is missing.

Round 2

Reviewer 2 Report

I have one minor follow-up comment:
In regard to the literature review, authors have not fully addressed the comments. Please conduct a review on latest works (5-10 journal articles) for the latest development of the research topic. Ensure the key elements, methodology, results, and limitations of the latest works are presented.
